# Natural Enemies of the Pear Psylla, *Cacopsylla pyri* (Hemiptera: Psyllidae), and the Possibilities for Its Biological Control: A Case Study Review in the Western Balkan Countries

Jovan Krndija [1,†], Aleksandar Ivezić [2,†], Ankica Sarajlić [3], Tijana Barošević [2,*], Boris Kuzmanović [4], Kristina Petrović [2,5], Isidora Stojačić [2] and Branislav Trudić [6]

1  Forecasting and Reporting Service in Crop Protection of Republic of Serbia, Agricultural Advisory and Professional Service Kragujevac, Cara Lazara 15, 34000 Kragujevac, Serbia; fitokragujevac@gmail.com
2  Biosense Institute, Dr. Zorana Đinđića 1, 21000 Novi Sad, Serbia; aleksandar.ivezic@biosense.rs (A.I.); kristina.petrovic@biosense.rs (K.P.); isidora.stojacic@biosense.rs (I.S.)
3  Faculty of Agrobiotechnical Sciences Osijek, Josip Juraj Strossmayer University of Osijek, Vladimira Preloga 1, 31000 Osijek, Croatia; ankica.sarajlic@fazos.hr
4  Faculty of Agriculture, University of Novi Sad, Trg Dositeja Obradovića 8, 21000 Novi Sad, Serbia; kuzmanovic.boris@gmail.com
5  Breeding Department, Maize Research Institute, Slobodana Bajića 1, 11185 Belgrade, Serbia
6  Faculty for Biology, University of Belgrade, Students' Square 16, 11158 Belgrade, Serbia; brauxsimple28@gmail.com
*  Correspondence: tijana.barosevic@biosense.rs
†  These authors contributed equally to this work.

**Abstract:** The accessible literature covered in this paper commonly highlights psyllids as a significant group of insects affecting pear trees, posing a continual challenge for commercial orchards. With the development of modern pear cultivation systems, *Cacopsylla pyri* Linnaeus 1758 (Hemiptera: Psyllidae) has emerged as a major pest in pear orchards across many European countries, including those in the Western Balkans. For years, the agricultural sector has primarily relied on chemical insecticides to control pear psyllas, but these methods often fail to produce satisfactory results. This is largely due to *C. pyri*'s rapid development of resistance to chemical treatments. Consequently, modern agriculture is increasingly shifting towards biological methods to manage *C. pyri*, involving the identification and conservation of its natural enemies. Although there is an abundance of research on the natural predators of *C. pyri* and their biocontrol applications across the globe, the Western Balkan region has conducted relatively few studies on the subject. Globally, various parasitoids, predators, and entomopathogenic fungi are often cited as effective against *C. pyri*. Specific species registered in the agroecological conditions of the Western Balkans include parasitic wasps such as *Trechnites insidiosus* Crawford, 1910 (Hymenoptera: Encyrtidae) and *Prionomitus mitratus* Dalman, 1820 (Hymenoptera: Encyrtidae), as well as the predatory bug *Anthocoris nemoralis* Fabricius, 1794 (Hemiptera: Anthocoridae). However, most Balkan countries have yet to fully utilise the potential of beneficial entomofauna or develop strategies for their commercial application at a national level. Considering that *C. pyri* is a major pest in pear cultivation and its natural enemies have not been thoroughly explored in most of the Western Balkans, this paper aims to review the literature data on available natural enemies of pear psyllas and to highlight and promote their undeniable potential in biological control.

**Keywords:** biological control; *Cacopsylla pyri*; natural enemies; parasitoids; pear; predators

## 1. Introduction

In Europe, the pear (*Pyrus communis* L.) is one of the most common domesticated fruit tree crop species in the temperate climate zone, cultivated on over 100,000 hectares and categorised as the second most prevalent pome fruit with an annual yield of over 2 million

tons [1,2]. Spain and Italy are the largest pear producers in Europe, accounting for 25–30% of European production, while Serbia, Bosnia and Herzegovina, and Montenegro are the most significant producers in the Western Balkans (WB) [3]. In Serbia, in addition to the apple (*Malus domestica* B.), the pear is the most prevalent and is produced over more than 7 thousand hectares of land, placing Serbia 10th in Europe in its production [4]. Bosnia and Herzegovina (BiH) are the second most important pear producer in the Western Balkans, with over 6 thousand hectares dedicated to pear cultivation [5]. Pear orchards in BiH rank third after the plum (*Prunus domestica* L.) and apple, characterised by a pronounced growth trend [6]. There is a long tradition of pear tree cultivation in northern Montenegro [7]. According to statistical data, Montenegro's pear fruit production in gardens and extensive orchards is 2648.9 tons (total production), with a yield of 10.4 t/ha [3]. Unlike Serbia, BiH, and Montenegro, the pear's share in the total fruit production in Croatia varies from year to year, both in intensive and extensive orchards, presenting around 750 hectares with an average yield of 3.4 t/ha. Interestingly, the area of organic pear orchards in Croatia is constantly increasing. In 2017, the total production was across 93 hectares of land, while in 2022, it grew to 221 hectares (https://tinyurl.com/ys4fe79k, accessed on 5 December 2023), suggesting that the number of natural enemies of *C. pyri* may also be increasing in these orchards. Like other Balkan countries, Albania and North Macedonia have a long tradition of pear cultivation, with a rising trend in pear production in Albania. Albanian pear production has been growing by an average of 2.5% year-on-year since 1966. In 2021, it totalled 15,130 metric tons, placing Albania 45th in the global ranking (https://www.reportlinker.com/clp/country/2862/726428, accessed on 7 December 2023).

Pear pests include various insects, with special attention given to psyllas, which pose a major challenge in commercial pear orchards. In Europe, economically significant psyllas species include the pear psylla, *Cacopsylla pyri* Linnaeus 1758 (Hemiptera: Psylidae), and the pear sucker, *Cacopsylla pyricola* Foerster, 1848 (Hemiptera: Psylidae) [8]. In the 1950s and 1960s, with the introduction of modern pear cultivation systems, the pear psylla (Figure 1) became the most important pear pest in many European countries, including those in the Western Balkans [9,10]. Damage from *C. pyri* can be direct or indirect. Larvae (Figure 2) cause direct damage by feeding on plant sap, reducing the leaf's assimilation surface, and secreting honeydew on which sooty mould fungi, *Capnodium* spp., develop, leading to a decrease in fruit quality and, therefore, a loss in market value. Additionally, the pear psylla transmits various phytoplasmas, such as *Candidatus Phytoplasma pyri* [11], which often induce a decline in pear production and represent one of the most devastating diseases on *Pyrus communis* in Europe [4]. Managing the presence and density of pear psyllas in agricultural orchards and natural tree populations is a demanding task for agronomists for several reasons: the species goes through a large number of reproductive cycles and produces a high number of individuals per generation, where, in most cases, there is also the occurrence of overlapping generations and the production of honeydew, which acts as a natural barrier to insecticides [4]. In the Western Balkans, broad-spectrum insecticides remain the most common method of controlling this pest, which reduces the number of natural enemies and often does not provide satisfactory results [12]. This is explained by the fact that *C. pyri* quickly develops resistance to the active ingredients, and the honeydew significantly protects the pest from the effects of applied insecticides [9]. Due to the emergence of resistance, the number of treatments during vegetation often increases, averaging from 5 to 10 times per vegetation [13]. Although chemical control measures are still the most common approach for controlling pear pests on a global scale, modern agriculture, primarily due to increasing consumer concerns about food safety and pesticide residues in final products, gives preference to integrated pest management (IPM) and, above all, biological pest control in pear orchards [14,15]. For these reasons, the scientific community and the general public are increasingly focusing on finding biocontrol solutions for *C. pyri*, which involve the identification of all the natural enemies of the pear psylla, studying their population dynamics and biological characteristics, as well as conserving the already present, beneficial entomofauna in pear plantations [16–19].

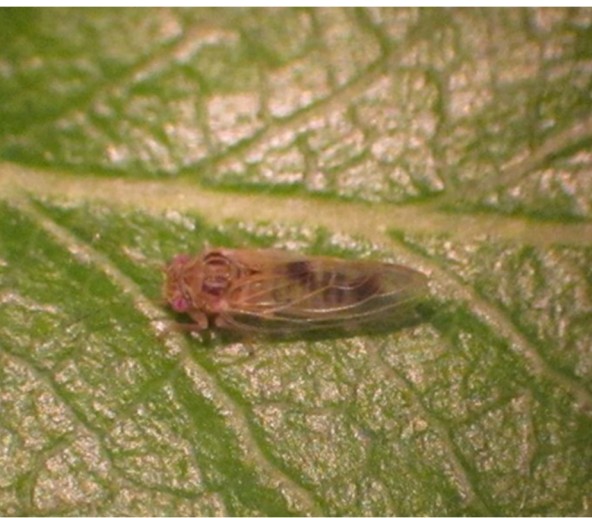

**Figure 1.** Adult of the pear psylla (photo: Krndija, J., July 2023 Serbia).

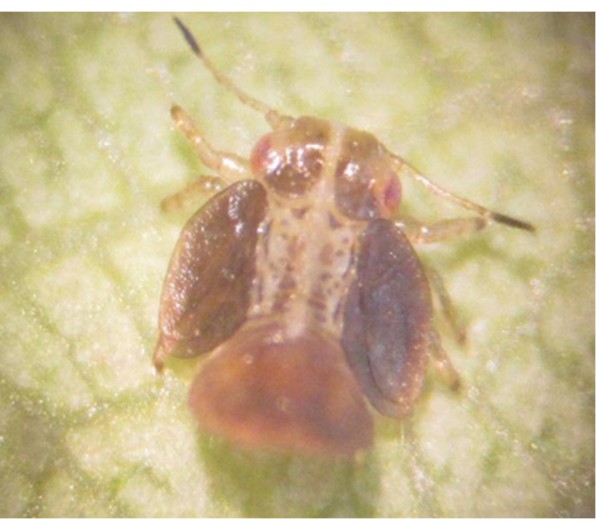

**Figure 2.** Larva of the pear psylla (photo: Krndija, J., July 2023, Vinča, Serbia).

Implementing biological control measures in pear protection programmes can significantly contribute to reducing the pear psylla population. One of the key strategies in the biocontrol of psyllas is undoubtedly the conservation of native populations of natural enemies, whose activity provides invaluable assistance in controlling pear psyllas [16–18]. However, the prevalence of natural enemies in commercial pear orchards is often insufficient to maintain the pest below the economic threshold, necessitating the introduction of laboratory-produced individuals and augmentation of the beneficial insect population [20,21]. To establish an optimal relationship between psyllas and their natural enemies, it is crucial to consider all beneficial organisms present in pear orchards in order to select the most appropriate species for biological control of the targeted pest [19]. To date, a large number of publications worldwide have listed the natural enemies of *C. pyri* with high potential for biological control of this pest. Among these, various species of parasitoids and predators are often mentioned, some of which have been registered in the Western Balkans, such as the parasitic wasps *Trechnites insidiosus* Crawford, 1910 (Syn. *Trechnites psyllae* Ruschka, 1923) and *Prionomitus mitratus* Dalman, 1820 (Hymenoptera: Encyrtidae), as well as the predatory bug *Anthocoris nemoralis* Fabricius, 1794 (Hemiptera: Anthocoridae) [10,22–24]. Although literature sources mention the sporadic commercial use of certain beneficial insect species in the biological control of pear psyllas in both Europe and the American continent [25,26], Western Balkan countries still do not adequately recognise

the autochthonous populations of natural enemies as valuable in biological control or evaluate their impact on controlling agricultural pests. The same trend is present in terms of conserving existing species or implementing certain species into national strategies for biological control of *C. pyri*, despite the significant damage caused by this pest in intensive pear production. Considering that *C. pyri* represents the most important pest in the modern system of pear production and that its natural enemies are still insufficiently investigated in most Western Balkan countries (Croatia, Bosnia and Herzegovina, Albania, Montenegro, and North Macedonia), the aim of this paper is to review regional (WB) and foreign literature data on the natural enemies of this pest and to highlight their potential contribution to the biological control of the pear psylla. Although the question of biological control of the pear psylla as a vector arises, since insufficient suppression of the vector can cause massive spread of the disease, this problem is not the subject of the paper. The focus is on the direct damage of pear psylla and its natural enemies discovered in the WB region as possible biocontrol agents.

## 2. Predators of the Pear Psylla

The most significant predators of pests in plant production can be divided into generalists (those that feed on various prey) and specialists (focused on a specific species), with specialists considered more effective in the biocontrol of pests [27,28]. Nevertheless, this classification can be seen as provisional in the case of the pear psylla, as the only insect species that has shown a marked affinity for *C. pyri* is the predatory bug *Anthocoris nemoralis*, which is most commonly used in pear psylla biological control programmes. Although *A. nemoralis* feeds on the pear psylla, it also preys on other insect species, such as the apple sucker *Cacopsylla mali* Schmidberger, 1836 (Hemiptera: Psyllidae) and Aphididae Latreille, 1802 (Hemiptera). Besides *A. nemoralis*, other members of the Anthocoridae, such as *Anthocoris nemorum* Linnaeus, 1761 (Hemiptera: Anthocoridae) and *Orius* spp., can have an impact on reducing the population of psyllas, but they are not as effective as *A. nemoralis*, which has proven to be the most efficient predator of *C. pyri* [18,21,29,30].

Natural enemies of the pear psylla include several different taxonomic groups, but they vary significantly in their potential as biological agents in controlling these pests. Among general predators, spiders, especially species from the Philodromidae (Araneae) and Clubionidae (Araneae) families, ants (Hymenoptera: Formicidae), Coccinellidae (Coleoptera), Chrysopidae (Neuroptera), and Miridae (Hemiptera) should be highlighted.

This paper will provide detailed descriptions of predator groups recorded in the Western Balkans, covering their relevance and potential success in the biological control of the pear psylla. Additionally, specific predators that were not recorded during inventories in the WB region but were documented in other parts of Europe will be considered. Their impact on the effectiveness of biological control can be positive, contributing significantly to the reduction in population of the mentioned pest, or negative, through various antagonistic interactions, competition, and predation over natural enemies. This analysis provides a better understanding of the complexity of biological control and its potential effects in different geographical contexts. To understand the climatic conditions in the Western Balkans, research covering the Palaearctic region is relevant, especially in agroecological conditions similar to those in this part of Europe. Therefore, the focus is on research conducted in Europe, with special emphasis on areas with temperate continental and Mediterranean climates. This is because the natural populations of natural enemies typical of Europe are also present in the Western Balkans.

According to a study conducted in Serbia [22], an inventory of natural enemies of the pear psylla was carried out from 2005 to 2009 at 167 sites across Serbia. During this period, 21 predators were recorded, with *A. nemoralis* present in the greatest number at investigated sites. Two other species, *Adalia bipunctata* Linnaeus, 1758 (Coleoptera: Coccinellidae) and *Coccinella septempunctata* Linnaeus, 1758 (Coleoptera: Coccinellidae), stood out as dominant species with continuous presence at sites and over the years of recording. However, there was evident spatial and temporal variability in the diversity of the present species. Data

from this study were complemented by research in 2013 [31] focusing on members of the order Hemiptera and suborder Heteroptera. In both papers, Jerinić-Prodanović and Protić highlight very significant data on the species assemblage of the entomofauna in pear orchards in Serbia, with the potential for biological control of pear psyllas. Additionally, the absence of spiders is notable, as they are cited in other regions of the world as very significant predators [16,18,32].

According to the available literature, a study in Croatia evaluated the effectiveness of various insecticides on *C. pyri* through integrated plant protection measures, where natural enemies were also monitored. Along with *C. pyri*, natural enemies from the families, Miridae, Nabidae, Coccinellidae, and Chrysopidae were regularly present, but in smaller numbers, on all inspected trees [33]. In the Republic of North Macedonia, there are no published data on this topic; however, attempts have been made to control pear psyllas by introducing *Orius laevigatus* Fieber, 1860 (Hemiptera: Anthocoridae) with partial success [34]. Entomophagous fauna was the subject of examination in the area of East Sarajevo in Bosnia and Herzegovina. At that time, members of the following predator families of the pear psylla were identified: Anthocoridae, Coccinellidae, Syrphidae (Diptera), and Chrysopidae [6,35].

## 3. Specialist Predators

The predatory bug *A. nemoralis* has been identified in different years of research in the agroecological conditions of Serbia during May (2006), June (2006, 2007), September (2005, 2009), and October (2006, 2008) [22]. During the 12-year research (1997–2008), *A. nemoralis* was also registered in Croatia as a predator of *C. pyri* [36]. According to research on psylla predators in Bosnia and Herzegovina, this bug was not recorded [6,35]. In Central Europe, it was most prevalent in the second half of the growing season, in July–August, when it had the greatest impact on reducing the pear psylla population, while its presence in spring was less frequent [18] and insufficient for satisfactory control of *C. pyri* during this time of year. The same results were obtained in a study in Türkiye [21]. The release of mass-reared predators early in the season does not yield favourable results, as shown in different parts of Europe [37]. This can be attributed to the high mortality rate and wide spatial distribution of predatory bugs [29,37]. The authors of [21] also cite factors such as the effects of broad-spectrum insecticides and the presence of ants, which negatively affect beneficial insects in psylla suppression.

In different ecological conditions, *A. nemoralis* can produce three to four generations per year [38]. Its natural range includes Europe and the Mediterranean region. The adult overwinters under tree bark and in similar hidden places, so it is ready to hunt as soon as prey becomes available and weather conditions permit [30]. It appears early in spring, when females lay eggs under the epidermis of leaves [38]. It is found on perennial plants, shrubs, and hedgerows surrounding orchards, from where it migrates into the plantations. In such vegetation, it feeds on various aphids and psyllas during spring before migrating to orchards [39]. The period of migration from spontaneous flora to cultivated crops is of great importance regarding the potential control of the common pear psylla. By that time, the common pear psylla may have already caused damage to orchards. This migration usually occurs during April and May, simultaneously with the increase in the psylla population. Trees from the genera Rhamnus, Laurus, and Pistacia provide shelter, where this species reproduces more quickly and successfully than on cultivated or wild pear trees. Later on, *A. nemoralis* migrates to nearby orchards for more effective control of pear psyllas [40]. Grass vegetation maintains populations of predatory bugs or attracts them [30], which is significant from the standpoint of conserving *C. pyri* predators. *A. nemoralis* feeds equally on both eggs (Figure 3) and larvae of *C. pyri*, preferring to feed on the fourth larval stage, and both the adult and larva of *A. nemoralis* exhibit predatory behaviour [30]. Pear psylla adults are less frequently preyed upon by predators due to their naturally increased mobility, allowing them to easily evade predators [39]. Larva feeds by extracting hemolymph from the prey's body with its piercing–sucking mouthparts [38].

The predator lays eggs more on pear leaves than on other fruit trees, such as apples [41], and prefers leaves with honeydew [42]. This could be related to the fact that larvae need an easily accessible food source after hatching, since the early larval stages of the pear psylla, which are typically covered in honeydew, are less mobile [43]. Additionally, it has a high preference for laying eggs on leaves without mechanical damage, and eggs are most often found far away from the leaf edge, in the central part of the leaflet [41].

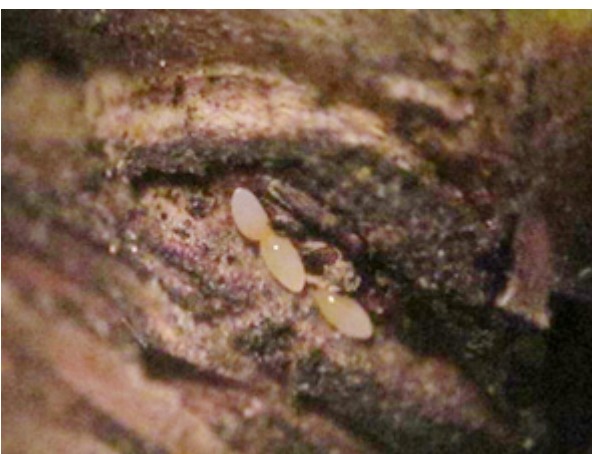

**Figure 3.** Eggs of *C. pyri* (photo: Krndija, J., Vinča, Serbia, February 2023).

When it comes to the mass rearing of predatory bugs and their introduction into orchards, there are methods for the laboratory cultivation of various species of the genus *Anthocoris*, including *A. nemoralis* [44]. This species reproduces faster than the most predatory insect species, developing from egg to adult in about fifteen days [38]. This is a trait that makes it a suitable and recommendable agent for biocontrol. The basic idea is to increase natural predator populations, thereby enhancing their effect on controlling the target organism, especially when the predator population in the natural environment is insufficient to keep the pest under control.

However, the success of such interventions crucially depends on several key factors: the number and developmental stage of predators used in the intervention, timely application, limited use of insecticides, and the presence and influence of other species interacting with the predators. The quantity recommended by specialised companies producing *A. nemoralis* as a biological agent, such as "Bioplanet", is usually higher than that used in experiments, ranging from 2000 to 3000 adult individuals per hectare, with two to three introductions, in spring when the air temperature exceeds 10 °C. The timing of the application needs to be positioned so that the prey, i.e., *C. pyri*, is present, but not in high numbers, which is precisely the spring period before *A. nemoralis* begins its mass migration from the vegetation surrounding the orchards. At the beginning of the growing season, weather conditions play a more significant role in reducing pear psylla populations than natural enemies [21], because predatory bugs are not yet present in sufficient numbers. The choice of the optimal developmental stage of *A. nemoralis* for mass release in the wild also arises. The adult stage is most commonly used, as it directly exhibits predatory behaviour on the pear psylla as well as its offspring [29]. Regarding the utilisation of *A. nemoralis* larvae as a biological agent, the advantage could lie in the fact that the rearing costs are lower than for adults, and the same applies to eggs [29]. However, the ability of larvae to disperse is limited compared to adults due to the reduced mobility of larval stages. Also, a greater number of larvae is needed than in the case of adults. Regarding the limited use of insecticides as a prerequisite for successful biological control, it should be noted that the population density of natural enemies of the pear psylla was higher in untreated than in treated orchards in spring and early summer, but the same stands for the pest itself, while this does not apply to late summer and autumn, since both predator and prey are mobile species and subsequently colonise the orchard regardless of previous treat-

ments [21]. The use of bioagents, such as those based on predatory bugs, combined with broad-spectrum insecticides is not recommended due to their extremely negative effect on beneficial insects as well. Insecticide use should be in accordance with guidelines for integrated pest management.

This requires the usage of newer generations of selective insecticides that have a satisfactory effect on controlling the pear psylla and are significantly less harmful to their predators [45]. In organic pear production, i.e., without the use of synthetic insecticides, *A. nemoralis* was the most significant predator of psyllas [46], making this production system most recommended for maximising the impact of the natural predator population. In specific research, this true bug was present when organophosphorus insecticides were excluded from the treatment programme, while it was not found in the year when this group of insecticides was used. The use of kaolin-based preparations as a repellent for the pear psylla is effective and safer for natural enemies [13,47].

A prerequisite for the successful application of *A. nemoralis* as a biological agent is the prior existence of populations of this predator in the targeted orchard and its surroundings in sufficient numbers, but this is not always the case [16]. If this condition is met, intervention should be carried out in spring, before the migration of the predatory bug into the orchard, when its population has not yet grown enough to reduce the local *C. pyri* population, and before it causes damage in the orchard. In this process, it is necessary to use chemicals rationally and to use only selective insecticides and/or alternative plant protection products.

The use of A. nemoralis as a biological control agent (BCA) is considered safe according to EPPO Standard PM 6/3 (5), concerning no adverse effects or with acceptable adverse effects. Some of the countries where it has been used include Belgium, Denmark, Germany, The Netherlands [48].

## 4. General Predators (Generalists)

The contribution of general (polyphagous) predators in suppressing pest populations can be significant, especially if they appear early in pest development, facilitating the action of specialised predators later in the season [28]. According to [49], general predators can be considered effective biocontrol agents if they meet the following criteria:

- They are present in sufficient numbers in the agroecosystem;
- They have low intraguild predation;
- Alternative prey is scarce in the agroecosystem, and pest population growth is slow.

These conditions are usually met during the winter and early spring in orchards. However, later in the season, when pest population density increases exponentially, general predators are unable to suppress the pest due to their inability to respond adequately to prey abundance [50]. Therefore, achieving complementarity between both types of predators is crucial for successful biological control, especially considering that field research shows that in approximately 75% of cases, generalist predators, as individual species or groups of species, are capable of significantly reducing pest numbers [28].

### 4.1. Spiders (Araneae)

Spiders are among the most numerous groups of general (generalist) predators found during inventories of arachnid natural enemies of pear psyllas in European pear orchards and are also among the most effective general predators [16,18,51]. Specific families such as Philodromidae (Araneae) and Clubionidae (Araneae) are prominent in Central Europe [18], while in southern Europe and the Mediterranean, important families include Philodromidae, Salticidae, Miturgidae, Oxyopidae, Theridiidae (Araneae), and genera such as *Philodromus* (Araneae: Philodromidae), *Oxyopes* (Araneae: Oxyopidae), *Cheiracanthium* (Araneae: Cheiracanthiidae), *Icius* (Araneae: Salticidae), and *Neoscona* (Araneae: Araneidae) [51]. In Serbia, there has been no inventory of spiders present in pear orchards, nor has their contribution to *C. pyri* control been evaluated, although research has been conducted in other agroecosystems, particularly in Vojvodina [52,53]. In Croatia, research on

this topic is also scarce, with the literature mentioning studies where the predatory mites *Typhlodromus pyri* Scheuten, 1857 were found in apple orchards [54], as well as various species of beneficial spiders in orchards and vineyards [55,56].

In Spain, the increase in spider populations during summer and autumn allows for effective control of *C. pyri* populations, reducing the number of overwintering adults and affecting the initial population increase in the next season. This spider activity increase at this time of the year is associated with a decrease in the number of ants and *Pilophorus gallicus* Remane, 1954 (Hemiptera: Miridae), which are the species that typically have an antagonistic effect on spiders [16]. Likewise, in Central Europe, Philodromidae and Clubionidae have been found to feed on pear psylla populations during spring and summer, but the presence of ants was not observed in significant numbers [18]. In fact, in this part of Europe, tree-dwelling spiders have been identified as the most important natural enemies of pear psylla, outnumbering Anthocoridae, which are considered specialised predators of this pest [18]. Spiders stand out for their activity during the winter months, when other natural enemies of pear psylla are inactive. Their ability to act on overwintering populations of these pests during winter and early spring is a key characteristic compared to other predators [32,57].

*Clubiona* spp. are very numerous in orchards [18,58] and are also active at low temperatures during winter, contributing to the reduction of overwintering psylla populations. The effectiveness of controlling pear psylla during winter increases with the density of winter spider populations [59]. Various strategies aim to enrich natural habitats by installing shelters for spiders to increase the number of individuals that successfully overwinter. Installing corrugated cardboard bands around trees and branches to provide additional overwintering sites for predatory spiders enhances predatory pressure on pear psylla populations and likely reduces intraguild predation among spiders sharing the same prey, in this case, pear psyllas [59].

A review of the available literature did not identify any attempts to use spiders as reared and released biological agents in controlling pear pests. In Croatia, most of the aforementioned spider species have been identified [60], but there is a lack of research on the impact of spiders on *C. pyri* in Croatian orchards. In the rest of the Western Balkan region, there is a lack of research and data about the presence of spiders in pear orchards and their impact on psyllas.

### 4.2. Ants (Formicidae)

Suppression of psyllas by ants has not been recorded in the Czech Republic [18]. Studies show that predatory ants can significantly reduce the population density of various psylla species [61], some of which pertain to Spain [16,62]. In Serbia, the presence of ants was not recorded during the inventory of *C. pyri* predators [22], which is consistent with other research conducted in Central Europe [18]. Research in other Western Balkan countries also does not report the presence of ants [35]. The abundance of ants increases with the presence of honeydew, but only while approximately 5% of the leaf surface is covered with honeydew, so washing off excess honeydew is recommended to enhance the predatory effect of ants on *C. pyri* [62]. The abundance of Coccinellidae and Anthocoridae is significantly lower in the presence of ants [21,62]. It should be noted that ants have an opportunistic relationship with Hemiptera populations, acting protectively towards these insects when taking honeydew as a carbohydrate source, and when their needs are met or they require prey as a protein source, they enter into a predatory relationship with pear psyllas [63]. No significant impact of pesticides on ants has been recorded, as there was no significant difference in ant abundance between treated and untreated orchards, most likely due to the lifestyle of ants, which avoids direct exposure to chemicals [16], and detoxification mechanisms, especially those characteristic of the queen [64].

### 4.3. Ladybugs (Coccinellidae)

Coccinellidae have been recorded in research conducted in Bosnia and Herzegovina [35], Croatia [33], and Serbia [22]. They are primarily predators of aphids, but adults and larvae of this family can also feed on psyllas, especially their eggs and younger larvae. Among them, the aforementioned *Adalia bipunctata* (adult) in Serbia was most commonly found during April and May, predominantly feeding on *C. pyri* eggs [22]. In Central Europe, an increased abundance of Coccinellidae was recorded in late spring, feeding on pear psyllas only if aphids, their main preferred prey, were not available [18]. Various species of Coccinellidae were also among the most common predators in research conducted in Turkey [21]. The absence of these predator species in research conducted in Spain is noticeable [16], which can be associated with the high abundance of ants and *P. gallicus* found at that time. Due to the lack of data and research in the WB region, we refer to the data from other parts of Europe. Both *A.bipunctata* and *C.septempunctata* are recommended by the EPPO as BCA for the control of aphids and have been used in a number of countries in the EPPO region [48].

### 4.4. Mirid Bugs (Miridae)

*Pilophorus gallicus* significantly reduces the number of pear psyllas in mid-spring in southern Spain because it overwinters as an egg under the barks of pear trees, and its larvae appear early in spring, reaching high numbers at the same time as psyllas, so this early emergence is certainly an advantage compared to *A. nemoralis* [16]. The limitation of this species as a biocontrol agent lies in the fact that spring populations depend on the oviposition of the previous autumn, the development of a small number of generations during the year, and the generalist predatory nature exhibited by this insect. Additionally, some *Pilophorus* species compete with predatory bugs (Staubli et al., 1992, cited in [16]), which can significantly affect the latter's success in biological control. In research on biological enemies in Türkiye, *Deraeocoris* spp. (Hemiptera: Miridae) were recorded [21], while Miridae were neither found in the Czech Republic [18] nor in Bosnia and Herzegovina [35]. During inspections in Serbia, Miridae species were found sporadically [22], as well as in Croatia [33], but it is important to consider them due to the complex and multifaceted roles they play in the ecosystem. Species of the Miridae family are generally phytophagous, but research has shown that a third of representatives of this family are zoophagous, and the genera *Macrolophus* and *Dicyphus* are commercially applied in protected environments [65]. According to the catalogue of Miridae bugs [66] in Croatia, 12 species from the genus *Dicyphus* and 3 species from the genus *Macrolophus* were identified, including the notable species *Campyloma verbasci* Meyer-Dür, 1843, which has been found to be a predator of *C. pyri* in other studies [67]. This predatory bug species was also confirmed in Serbia on *C. pyri* [23]. In Croatia, the species *Pilophorus perplexus* Douglas and Scott, 1875 from the Miridae family and *Orius niger* Wolff, 1811 from the Anthocoridae family were also identified as predators of the pear psylla [36].

### 4.5. Common Flower Bug (Anthocoris Nemorum)

*Anthocoris nemorum* has been found in Serbia [22,23] and Bosnia and Herzegovina [35]. In Serbia, it has been recorded during May–June [22,23]. It was also detected as a predator of pear psyllas during research in other parts of Europe, for instance, France and Denmark [29,68]. This predator has a more polyphagous nature compared to *A. nemoralis*, preferring to feed on aphids rather than psyllas [29]. In Croatia, this species has been identified on apples as a predator of aphids [69]. It is recognized by the EPPO as safe for the environment and native species with *C. pyri* and thrips as the main target pests. A number of countries where this species is used as a BCA include Belgium, Denmark, France, Italy, Jersey, Netherlands, and the UK [48].

*4.6. Green Lacewings (Chrysopidae)*

Chrysopidae are present in all Western Balkan countries where research on the natural enemies of pear psyllas are available [6,22,35]. In Serbia, the following species were identified: *Chrisopa pallens* Rambur, 1838, *Chrysoperla carnea* Stephens, 1836, and *Chrisopa* sp. (Neuroptera: Chrysopidae), found during May–July and in November [22]. Species of this family have also been recorded in other parts of Europe [21,68]. Of the above species, EPPO also considers C.carnea to be acceptable from a safety point of view and can therefore be released against its main target pests. The use of this species as a BCA has been recorded in Austria, Belgium, the Czech Republic, Denmark, Finland, France, Germany, Greece, Guernsey, Ireland, Italy, the Netherlands, Portugal, Spain, Sweden, Switzerland, and Great Britain [48].

## 5. Parasitoids of the Pear Psylla

Similar to predators, among the parasitoids of pear psylla, there are certain species categorised as general parasitoids, as they parasitise a larger number of hosts, while many others are specialised, i.e., species that are highly specific to their host [19]. The authors of [19] emphasise that the commercial application of specialised parasitoids represents a promising alternative to the use of general ones, mainly due to their pronounced specificity towards a particular host, high host-finding ability, high fecundity, and reduced negative effect on the environment and local biodiversity. The fauna of pear psylla parasitoids in Europe is quite diverse, and among the registered species are *T. insidiosus*, *P. mitratus*, *Prionomitus tiliaris* Dalman 1820 (Hymenoptera: Encyrtidae), *Psyllaephagus procerus* Marcet 1921 (Hymenoptera: Encyrtidae), *Syrphophagus ariantes* Walker, 1837 (Hymenoptera: Encyrtidae), *Syrphophagus taeniatus* Förster, 1861 (Hymenoptera: Encyrtidae), and genera *Tamarixia* (Hymenoptera: Eulophidae) and *Endopsylla* spp. (Diptera: Cecidomyiidae) [21–23,26,70,71]. Despite a significant number of registered parasitoids of *C. pyri*, there are very few attempts at their commercial application and laboratory production, and even less information about their biological characteristics and ecological significance [1]. Of the aforementioned species, *T. insidiosus* is often underlined as the most abundant species in European pear orchards [1,16,19,21,70–73]. The literature data indicate the widespread distribution of this species, despite being quite susceptible to hyperparasitism [16,70,71,74] and extremely sensitive to chemical control measures, particularly broad-spectrum insecticides [16,75,76]. *Trechnites insidiosus* belongs to the group of endoparasites and is native to the Euro-Asian area, having been introduced to North America during the 19th and 20th centuries [1]. In 1965, this species was commercially applied in California as a biological agent to suppress a previously introduced population of psyllas [77]. Unfortunately, precise data on the population dynamics of the introduced parasitoids and their establishment in the treated area are lacking [19]. Despite a limited number of studies, *T. insidiosus* is still considered an extremely suitable agent for controlling psyllas, with great potential for integrated pear protection. Its activity is from April to late November, indicating its ability to be active even at lower temperatures, and this is particularly significant for its further commercial usage [70–73,78]. The first generation of *T. insidiosus* is not susceptible to hyperparasitism [66,68] and is active in the initial phases of pear development when no other active parasitoids or predators are present [19]. Several studies indicate an extremely high level of parasitism by *T. insidiosus*, which, according to various authors, ranges from 30 to even 100% [21,73,79,80], highlighting the high efficiency of this parasitoid in controlling psylla populations [81]. Despite its pronounced potential and favourable biological characteristics for commercial application, only a few laboratory studies have been conducted on the behaviour of *T. insidiosus* in parasitising the pear psylla [19]. To understand the interaction between *C. pyri* and *T. insidiosus*, [19] conducted a laboratory study examining the influence of the larval development of pear psylla on the quality of emerging parasitoid individuals. It was established that female *T. insidiosus* oviposit on all larval stages of pear psylla, but they prefer the older larval stages of the host, which are nutritionally richer for the development of the parasitoid. The same authors highlight the greater biological potential of emerging

*T. insidiosus* individuals in the case of parasitising older larval stages compared to the initial stages of *C. pyri* larval development [19].

In Western Balkan countries, there is a very small number of studies on parasitoids of *C. pyri*, with published results dating from a period when modern procedures for precise species identification, such as molecular tools and techniques, did not exist. Given the difficulty in morphological identification of certain parasitoids, primarily due to small body sizes and variability of morphological characteristics, species identification data obtained before the implementation of molecular markers in regular entomological practice should be taken with caution. This refers primarily to occasional errors in older literary sources dealing with the determination of parasitoid species and their taxonomic affiliation. A good example is parasitoids of the genus *Trichogramma* Westwood, 1833 (Hymenoptera: Trichogrammatidae), where molecular identification has allowed significant progress in the taxonomy of these organisms, as molecular analyses have revealed numerous errors in the systematics and identification of these insects [82,83]. In the Western Balkans region, with respect to inventories of the parasitoid complex of pear psylla, the most significant progress was made in Serbia, where the identification of the most important parasitoids of psyllas began in the 1970s. Studying the prevalence of psyllas in Serbia, [84] identified the presence of eleven species of parasitic wasps from seven families. In Vojvodina (Serbia), [10] studied the prevalence of natural enemies of *C. pyri* and *Cacopsylla pyrisuga* Foerster, 1848 (Hemiptera: Psyllidae), identifying the presence of four species of parasitic wasps: *P. mitratus*, *T. insidiosus*, *S. teaniatus*, and *Coccophagus lycimnia* Walker, 1839 (Hymenoptera: Aphelinidae). The same authors recorded four hyperparasitoids: *Marietta picta* Andre, 1878 (Hymenoptera: Aphelinidae), *Pachyneuron aphidis* Bouche, 1834 (Hymenoptera: Pteromalidae), *Pachyneuron concolor* Forster, 1841 (Hymenoptera: Pteromalidae), and representatives of *Charips* spp. (Hymenoptera: Cinipidae), which were not identified at the species level. The authors of [22], conducting systematic monitoring of pear psylla at 167 different locations in Serbia, recorded the presence of the following parasitoids: *P. mitratus*, *P. procerus*, *S. ariantes*, *S. taeniatus*, and representatives of the genus *Tamarixia*. This was the first report of *P. procerus*, *S. ariantes*, and representatives of *Tamarixia* spp. as parasitoids of *C. pyri*, and as a new species in the entomofauna of the Western Balkans [22]. Despite the fact that *T. insidiosus* is mentioned in various literary sources as the predominant parasitoid of *C. pyri* in Europe [16,21,73], in the region of the Western Balkans, this species was recorded only in Serbia [10,23]. The research in [22] indicates that *P. mitratus* is the most abundant parasitoid of the pear psylla in Serbia, as this species comprised half of the total number of collected parasitoids during the monitoring. The presence of *P. mitratus* was recorded from the first half of June to mid-November, but it was found in greater numbers in the second half of June [22]. In Europe, the species *P. mitratus* has been studied by several authors, who underlined it as a very significant natural enemy of various species of psyllas, primarily *C. pyri* [68,71,85]. In France, [68] determined that *P. mitratus* parasitises *C. pyri*, *C. pyrisuga*, and *Cacopsylla melanoneura* Foerster, 1848 (Hemiptera: Psyllidae), while the authors of [71] state that *P. mitratus* mainly occurs as a parasitoid of the first generation of pear psylla, and in subsequent generations it appears in slightly lower numbers compared to other parasitoids. *Prionomitus mitratus* is a polyphagous species that parasitises several species of psyllas from the genus *Arytaina* (Hemiptera: Psylidae), *Pexopsylla* (Hemiptera: Psylidae), *Cacopsylla* (Hemiptera: Psylidae), and *Trioza* (Hemiptera: Triozidae) [86]. Like *T. insidiosus*, *P. mitratus* was introduced to North America from Europe for the biological control of pear psyllas. However, satisfactory results were not achieved, as the species showed partial effectiveness in controlling psyllas [87]. In Serbia, [10] identified *P. mitratus* on *C. pyrisuga*, while the authors of [22] were the first to record this species on the larvae of *C. pyri* in Serbia. The second most prevalent parasitoid of pear psylla, according to [22], is *P. procerus*, which was recorded from the first half of June to mid-November, but in slightly greater numbers in mid-June. The same authors emphasise that *P. procerus* constituted 43.55% of the total collected individuals of *C. pyri* parasitoids. Another interesting finding relates to the species *S. ariantes*, which [22] recorded for the first time as a parasitoid of

*C. pyri* worldwide. Until then, *S. ariantes* was categorised only as a parasitoid of *Trioza urticae* Linnaeus, 1758 (Hemiptera: Triozidae) [86]. This was also the first finding of *S. ariantes* in Serbia. *S. taeniatus* was recorded in a significantly smaller percentage, which some authors consider a primary parasitoid [86], while others list it as a secondary parasitoid of *C. pyri* [68].

The authors of [23] conducted another extensive research study on psylla parasitoids at 51 sites from 2003 to 2017, identifying 27 different species of parasitoids, among which *P. mitratus* stood out as the most abundant parasitoid of pear psyllas in Serbia. Its predominant presence highlights this species as a potential candidate for inclusion in the national programme for biological control of *C. pyri*.

In other Western Balkan countries, there are almost no data on the presence of determined parasitoids of the pear psylla. Research conducted in Bosnia and Herzegovina in [35] indicates the presence of the aforementioned psylla predators, while parasitoids were not the subject of the research. The same applies to the Republic of North Macedonia, Albania, Montenegro, and Croatia, where there is a lack of literature data on parasitoids of *C. pyri* as well as information on the commercial application of parasitoids for controlling psyllas [34,88]. Although the data are scarce, we succeeded in selecting and presenting the geographical distribution (across Europe) of five key predators of pear psylla (Figure 4) according to the literature data [16,18,21–23,31,35,36,70,72,73,89,90].

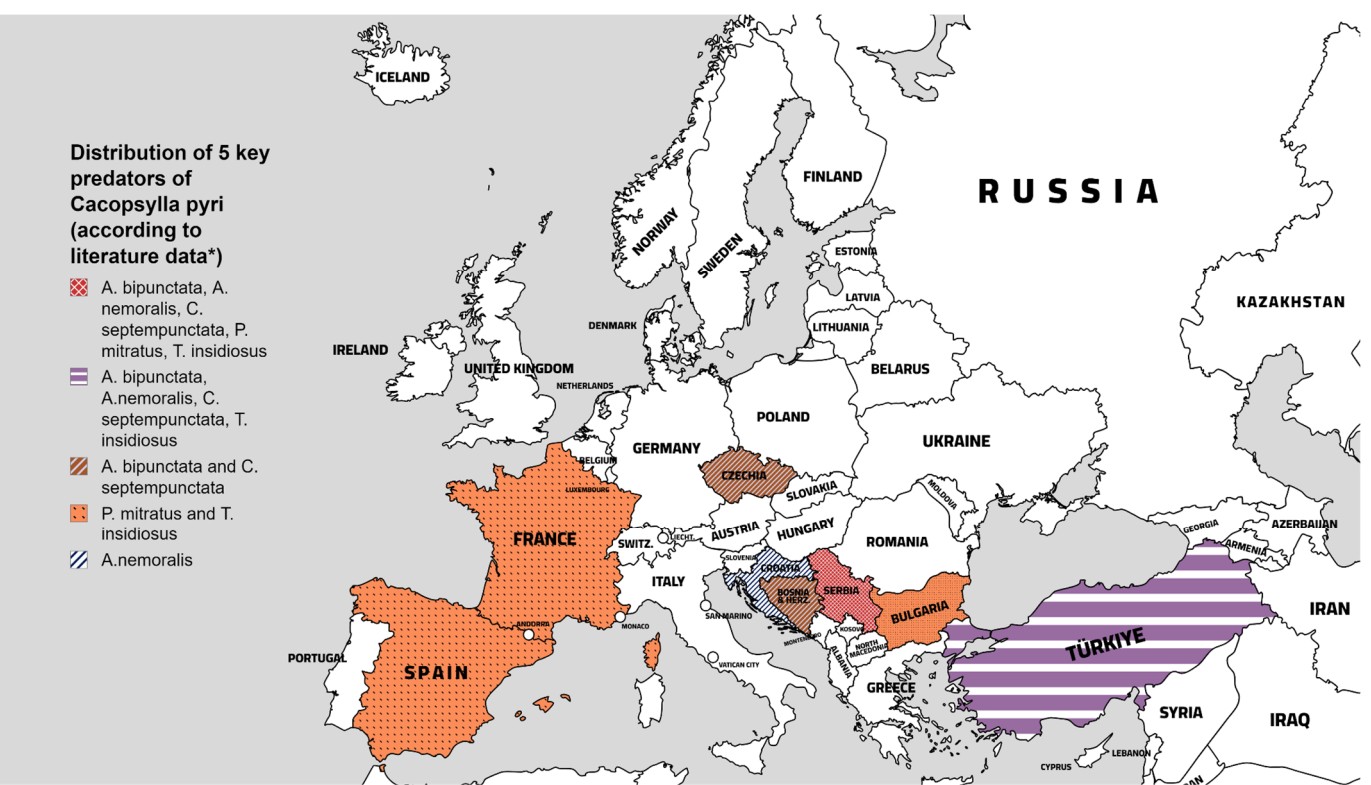

**Figure 4.** European distribution of five key natural enemies of pear psylla. The literature data * (although limited in scope) refer to southern Europe as a significant pool of natural populations of those predators and important for future breeding programmes of targeted organisms for biocontrol purposes. By far, Serbia is the only WB country with all five key predators identified, with Turkey and BiH as forth runners. Figure was created using the MapChart.net online tool (https://www.mapchart.net/europe.html (accessed on 27 February 2024)).

A list of natural enemies of *C. pyri* which could be monitored in the WB region based on recent inventory surveys conducted in certain countries of the region and other parts of Europe with similar agro ecological conditions is shown in Table S1 (Supplementary Files)

as guidelines for potential use in biocontrol actions against this pest. Therefore, the data in the table are supplemented with the rate of predation/parasitism where available, as well as the level of population abundance. For some organisms, only genus or family is represented, as the researchers did not determine the specimen to the species level.

## 6. Alternative Biological Solutions for Controlling *C. pyri*

### 6.1. Entomopathogenic Fungi as Biological Agents for Controlling *C. pyri*

Entomopathogenic fungi were once considered impractical agents for biocontrol of harmful insects in growing fruit due to the unacceptable level of damage to plants that occurs before the fungi begin to act [91]. However, the constant use of insecticides has led to the development of resistance in many key orchard pests. This problem, combined with public pressure to reduce pesticide use, has reignited research into natural microbiological agents for controlling harmful insects, including entomopathogenic fungi [92]. One of the major challenges in using entomopathogenic fungi is their interaction with pesticides, primarily insecticides and fungicides. Researchers have discovered that the sensitivity of entomopathogenic fungi depends on the type of fungus, strain, and type of applied fungicide. Therefore, it is necessary to test the compatibility of fungicides with selected entomopathogenic fungi [93,94]. The use of entomopathogenic fungi represents an exceptionally effective method of biological protection. Species from the genus *Fusarium* are mostly phytopathogenic, producing secondary metabolites, including mycotoxins, which are harmful to human and animal health. However, some species have shown entomopathogenic properties with a high insect mortality rate of over 90%. To date, effects on insects from the orders Blattodea, Coleoptera, Diptera, Hemiptera, Hymenoptera, Lepidoptera, Orthoptera, and Thysanoptera have been identified, making them potential tools for studying new insect species [95,96].

The nature of the damage caused by the pear psylla allows sufficient time for spores of entomopathogenic fungi to germinate, infect, and control the pear psylla. There is a possibility of introducing entomopathogenic fungi into the orchard ecosystem to provide long-term protection against this pest [92]. During the 1990s, the first species of fungi showing pathogenic effects on pear psylla were identified: *Beauveria bassiana*, *Paecilomyces fumosoroseus*, *Verticillium lecanii*, *Metarhizium anisopliae*, and *Metarhizium flavoviride*. The authors of [92] studied these entomopathogenic fungi under laboratory and field conditions. In the laboratory, *B. bassiana*, *P. fumosoroseus*, and *V. lecanii* caused close to 100% mortality of pear psylla nymphs seven days after its application to leaves. The efficacy of *Metarhizium* species was 40–50% nymph mortality. Based on laboratory tests, isolates of *B. bassiana* and *P. fumosoroseus* were approved for field trials. Three different formulations based on the conidia of these fungi were made and applied to pear trees infested with nymphs of pear psylla. After one application, nymph mortality ranged from 18.2% to 37.1%, depending on the type of formulation. The conidial formulation with acrylic polymer in water caused significantly higher nymph mortality compared to the formulation with water or a combination of water and paraffin oil. In addition to these formulations, two commercial bioinsecticides based on *B. bassiana* were tested: Naturalis L (Fermone Corp.) and GHA (Mycotech). After the application of Naturalis L, nymph mortality was 34.1%, while after the application of GHA, nymph mortality was significantly lower, 10.8% [92].

Recent research focuses on studying entomopathogenic fungi from the order Entomophthorales and their pathogenic effects on pear psylla [97]. Within this order, the subfamily Erynioideae stands out, which includes the genera *Erynia, Pandora*, and *Furia* [98,99]. Species from these genera are characterised by oval or pear-shaped conidia formed on dichotomously branched conidiophores. While infections by entomopathogenic fungi are very common in species from the family Aphidoidea, there are fewer recorded infections in species from the superfamily Psylloidea. To date, 15 species from the order Entomophthorales have been described. Despite their potential, no biological control agent based on these fungi has been commercialised yet, due to difficulties in their cultivation and inconsistent results in field trials [100]. In 2016, a new entomopathogenic

species, *Pandora cacopsyllae*, was isolated from infected individuals of *C. pyri* collected in an orchard in Denmark [101,102]. The conidia of this species are almond-shaped, and the conidiophores are dichotomously branched. The life cycle of this fungal species is still not well understood. Therefore, research is needed on its distribution, epidemiology, and overwintering. In laboratory conditions, *Pandora cacopsyllae* caused 89% mortality of adult pear psyllas [103]. Since this species has only been tested in laboratory conditions so far, its potential for biological control in field conditions needs to be examined with the aim of developing a commercial product. Converting a species from the genus *Pandora* into a biological control agent requires a formulation that ensures the virulence of the fungus when applied against the target insect. The formulation of a biological agent with a carrier material can improve its applicability, shelf life, growth, duration of sporulation, and intensity after field application. Biocompatible and biodegradable calcium alginate is a potential carrier for the encapsulation of species from the genus *Pandora* [103]. In addition to the carrier material, the formulation of a biological agent may also contain beneficial additives in the form of nutrients that allow the fungus to proliferate after field application, attractants for host insects, substances to protect the formulation from desiccation and UV radiation, or components that enhance the virulence of the fungus [104]. However, further research is needed to develop new mycoinsecticides in orchards, focusing on application methods (spore concentration, timing of application), and studying compatibility with other pesticides, especially fungicides.

### 6.2. Kaolin as a Non-Chemical Alternative in Controlling C. pyri

Kaolin is an aluminosilicate mineral, white in colour, non-abrasive, and soluble in water. It creates a barrier on plants when sprayed [105], acting on a large number of pests in fruit production either repellently or by reducing oviposition, and can even cause death in insects directly exposed to clay particles [106]. Research on the effectiveness of kaolin on a large number of harmful insects has already been published. Effectiveness varies depending on the pest species, weather conditions, applied substance, etc. In the study by [107], almost 100% effectiveness of kaolin was found in controlling *C. pyri* on larvae and eggs of the overwintering generation, with no side effects on plants in terms of phytotoxicity. During 2017 and 2018, research was conducted in Croatia on the impact of kaolin clay on the mortality of *C. pyri* compared to other chemical insecticides. Kaolin clay (Cutisan) was applied at a dose of 35,000 g·ha$^{-1}$. The effectiveness of the treatment depended greatly on weather conditions and ranged from 37% in 2018 to 71% in 2017. In 2017, when the high effectiveness of kaolin clay was achieved, the results were very similar to those of chemical insecticides in terms of the number of eggs and different stages of larvae, while in 2018, these values were statistically significantly lower for kaolin clay [13]. This research concluded that kaolin clay should not be used alone, but its application within an integrated pest management practice yields very good results. According to numerous studies, kaolin is not considered toxic to insects but acts as a repellent, and it has no selectivity, as demonstrated in the study by [24], where a greater number of natural enemies were recorded in the control compared to the kaolin treatment.

### 6.3. Botanical Insecticides

Botanical insecticides are derived from various plant parts and have long been used in plant protection against pests. They can have contact and inhibitory activities and are considered growth and development regulators. Secondary plant metabolites such as terpenes, alkaloids, and phenols have various insecticidal effects, such as toxicity, repellency, feeding disruption, etc., [108]. Plant extracts usually have low toxicity and multiple effects, making them safer for non-target organisms. A significant advantage is that their effectiveness does not depend solely on one active component but on a range of related active substances, which gives them an advantage in terms of reducing resistance [109]. In the study by [110], the ovicidal and larvicidal effects of AkseBio2 (a product based on aromatic plants and essential and edible oils) for controlling pear psylla were tested. A strong repellent effect

was found in egg laying, as well as high mortality of eggs and various stages of larvae within 48 h. The highest mortality was recorded in the first larval stages (87.4%), while lower mortality rates (62.1%) were found in later larval stages at a concentration of 0.1%. Importantly, the study examined the impact on beneficial organisms, where no significant negative effect was recorded over seven days, nor was phytotoxicity observed, suggesting that the product can be successfully used as part of an IPM program. There are many studies on the impact of various plant extracts on different insect species, but the impact on pear psylla is generally poorly investigated. The effect of plant extracts depends on the plant species, solvent, insect species, insect stage, etc. Adult insect stages are often highly resistant to plant extracts [111,112], and the best results are obtained in the first larval stages [113,114], although high effectiveness in adults has been demonstrated for psyllas [115].

**7. Conclusions**

Given that the data on the most recent inventory of natural enemies of pear psylla in most Western Balkan countries are quite scarce and incomplete and that biological control of the most economically significant pear pest is largely neglected, it is necessary to complete knowledge of the entomofauna of beneficial insects in pear orchards in this part of Europe and to make an effort to promote biological protection measures. Considering that chemical control measures are still the most common method of controlling harmful organisms in the Western Balkans region, any effort aimed at promoting and implementing environmentally friendly solutions in regular agricultural practice is more than justified. Promoting integrated plant management, especially biological control measures, should be directed to all participants in agricultural production to raise the awareness of the general agricultural public to an appropriate level, thereby ensuring the production of healthier food, contributing to environmental protection, and preserving local biodiversity. As pear production in most Western Balkan countries is on the rise, national strategies should primarily be directed towards encouraging ecological plantations of this fruit species. An example of good practice is the Republic of Croatia, where there is a constant increase in organic pear orchards, which can be explained by aligning the national strategy with the principles and protocols of the European Union (the European Green Deal).

The literature data indicate that specialised natural enemies are more effective in controlling the pear psylla than general ones, but the contribution of general predators and parasitoids in reducing the population of this pest can be significant. Various activities are necessary to achieve compatibility and complementarity of specialists and generalists in the context of agroecological conditions for successful biocontrol, whether it relies exclusively on natural populations of *C. pyri* enemies in the orchard or on intervention by releasing mass-reared organisms to enhance and increase the effect of native populations. Also, in most Western Balkan countries, natural enemies of pests are neither included in the practice of long-term monitoring by agricultural expert services, nor are agronomists employed in such services trained to monitor them. Strategic plans and documents do not contain information about such activities, nor do they refer to these species. These local experts are mostly focused exclusively on harmful organisms, while the activity of beneficial insects is largely ignored. An example of good practice in Serbia is the monitoring of *Trichogramma* spp. species carried out by the Forecasting and Reporting Service for Plant Protection of Serbia (http://www.pisvojvodina.com accesed on 5 February 2024). The inventory of the entire genus was conducted through the system of this service [83,116,117], which is not part of the academic environment. It is important to note that all the studies mentioned in this work refer exclusively to the activities of scientific research organisations, and research should be expanded to include monitoring of beneficial organisms by these professional services, which are part of the agricultural state sector.

The European Union encourages and supports the implementation of IPM practices, especially biocontrol measures, as a crucial element in achieving sustainable agriculture goals at the global level. Until now, through various programmatic funds, such as Horizon

2020, the EU has contributed significantly to research and innovation activities that promote ecological and green agriculture (Table 1). This funding aims to enhance eco-friendly pest management methods, reduce pesticide use, and endorse a holistic approach that promotes crop health, environmental protection, and the long-term sustainability of agricultural production. Given that Western Balkan countries have access to EU funds, they have the opportunity to facilitate the establishment of such methods at the national level and implement certain non-pesticide solutions, such as biocontrol measures, in their long-term national strategies.

**Table 1.** Implemented HORIZON2020 and Cost Action projects over the past decade focused on IPM practices and eco-friendly solutions in agriculture.

| Project Name/Acronym | Keywords | Start Date/End Date | EU Contribution in Euros Per Project | Source |
|---|---|---|---|---|
| **HORIZON2020 projects** | | | | |
| Stepping-up IPM decision support for crop protection/IPM Decisions | IPM, crop protection, pest management, DSS, computer and information science, agro-meteorological network | 1 June 2019–31 May 2024 | 4,998,096.19 | https://cordis.europa.eu/project/id/817617 (accessed on 1 February 2024) |
| An EU-wide farm network demonstrating and promoting cost-effective IPM strategies/IPMworks | Agroecology, advisors, pesticides, holistic, sustainability, farmers, peer-to-peer | 1 October 2020–30 September 2024 | 6,000,005.00 | https://cordis.europa.eu/project/id/101000339 (accessed on 1 February 2024) |
| EcoStack | Biocontrol agents, barcoding, plant defence priming, sustainability, interaction | 10 September 2020–9 March 2024 | 9,963,866.00 | https://cordis.europa.eu/project/id/773554 (accessed on 1 February 2024) |
| Optimised Pest Integrated Management to precisely detect and control plant diseases in perennial crops and open-field vegetables/OPTIMA | IPM DSS, prediction models, early diseases detection, smart precision spraying technologies | 1 August 2018–30 June 2022 | 3,425,600.00 | https://cordis.europa.eu/project/id/773718 (accessed on 1 February 2024) |
| Agri and food waste valorisation co-ops based on flexible multi-feedstocks biorefinery processing technologies for new high added value applications/AgriMax | Agriculture, pilot plant, food, packaging, multi-feedstock biorefinery, agricultural and food processing waste valorization | 1 October 2016–30 September 2021 | 12,484,461.46 | https://cordis.europa.eu/project/id/720719 (accessed on 1 February 2024) |
| Trapview—Automated pest-monitoring system for sustainable growing with optimal insecticide use/Trapview | Sustainable agriculture, resource-efficient eco-innovations in agriculture, automated pest insect monitoring, insecticide spraying optimization, statistical forecasting models | 1 September 2016–31 September 2018 | 1,141,350.00 | https://cordis.europa.eu/project/id/733979 (accessed on 1 February 2024) |
| Coordinated Integrated Pest Management in Europe/C-IPM | IPM, pesticides, sustainability, plant protection, agriculture | 1 December 2014–31 December 2016 | 1,998,215.00 | https://cordis.europa.eu/project/id/618110 (accessed on 1 February 2024) |
| Innovative biological products for soil pest control/INBIOSOIL | Pesticide reduce, agriculture, biocontrol agents, sustainability | 1 July 2012–31 December 2015 | 4,984,654.20 | https://cordis.europa.eu/project/id/282767/it (accessed on 1 February 2024) |
| **Cost Action projects** | | | | |
| CA21134—Towards zer0 Pesticide AGRIculture: European Network for sustainability (T0P-AGRI-Network) | chemical pesticides, agroecology, sustainable agriculture, crop protection, transition | 19 September 2022–18 September 2026 | n.a. | https://www.cost.eu/actions/CA21134 (accessed on 25 January 2024) |
| FA1405—Using three-way interactions between plants, microbes and arthropods to enhance crop protection and production | Plant–arthropod–microorganism interactions, pest and disease management, plant growth and defence promoting microorganism, plant production | 10 March 2015–9 March 2019 | n.a. | https://www.cost.eu/actions/FA1405 (accessed on 25 January 2024) |
| FA1104—Sustainable production of high-quality cherries for the European market | sweet and sour cherry, rootstocks, climate change, sustainable agriculture | 16 April 2012–15 April 2016 | n.a. | https://www.cost.eu/actions/FA1104 (accessed on 25 January 2024) |
| 849—Parasitic Plant Management in Sustainable Agriculture | Agriculture, biocontrol, parasitic plants | 22 March 2001–21 September 2006 | n.a. | https://www.cost.eu/actions/849 (accessed on 25 January 2024) |

Finally, it is important to highlight key recommendations for the agricultural and protection sectors and the improvement of strategic monitoring practices of *C. pyri* and its natural enemies in local expert services in the Western Balkan countries (Table 2).

**Table 2.** Problem-solution matrix for future *C. pyri* biocontrol research in Western Balkan countries.

| Problems | Solutions |
|---|---|
| P.1.1. Lack of inventory of the predators/parasitoid complex of *C. pyri* in most Western Balkan countries, resulting in P.1.2. a lack of (any) recent data on the diversity of pear psylla predators/parasitoids. | S.1.1. Conducting detailed field research to inventory the predators/parasitoid complex of *C. pyri* in Western Balkan countries. S.1.2. Also, supporting and encouraging research projects focused on this topic will result in new data on the biodiversity of pear psylla predators/parasitoids. S.1.3. Determining the precise taxonomic affiliation and population genetic diversity of *C. pyri* and its natural enemies in Western Balkan countries using available molecular and biochemical markers. |
| P.2.1. A thorough understanding of all relevant ecological factors at the level of each agroecosystem/landscape is necessary for the successful application of biological control of the common pear psylla on pears in Western Balkan countries, which involves an analysis of climate, geographical characteristics, the presence of other insects or predators, as well as the impact of agricultural practices on the diversity of this pest insect. | S.2.1. Conducting detailed research that would cover all ecological factors specific to each agroecosystem/landscape where pears are grown (domesticated or wild types) in Western Balkan countries. S.2.2. Integrating these studies into interdisciplinary collaborative regional and cross-border projects, including EU HORIZON2020 and COST Action programmes, among others. S.2.3. Integrating proven practices that benefit the survival and development of pear psylla natural enemies (i.e., planting Rhamnus and Laurus trees around pear orchards and maintaining grass vegetation, which both attract *A. nemoralis*, or installing cardboard belts on pear tree trunks and branches in order to enhance the overwintering of spiders). |
| P.3.1. Insufficient information on the presence of biodiversity among the natural enemies of *C. pyri* and the degree of predation and parasitism for each of them. | S.3.1. A more detailed inventory of all significant biological enemies of *C. pyri* within the entire WB region, including climate data modelling and species distribution predictions. S.3.2. Determining the degree of predation and parasitism for each species to understand the available biological potential and to set a basis for further research and application actions. |
| P.4.1. Lack of research involving experimental application of commercial predators/parasitoids, i.e., laboratory-bred individuals, to evaluate their effectiveness and field performance in controlling the common pear psylla. | S.4.1. Piloting local biocontrol experiments over *C. pyri* using drones and other available robotic technologies through the aforementioned cross-border research initiatives. S.4.2. Developing a study on the economic cost–benefit analysis of using the aforementioned biocontrol technologies in regular agricultural practice compared to other available chemical and non-chemical treatments. |
| P.5.1. EU candidate and non-candidate countries of Western Balkans (Albania, Bosnia and Herzegovina, Kosovo *, Montenegro, North Macedonia, Serbia) national agricultural policies are still not aligned enough and effectively with relevant EU Acquis Communautaire and negotiation chapters' recommendations (chapter 11). Some of the examples of poor agricultural rule of law in the mentioned countries that need to be changed and intervened upon are corrupted government structures and decision makers, a lack of political will for EU-based reforms in the area of agriculture, unreformed national policies regulating local realities that damage the environment and IPM, etc. | S.5.1. Relevant stakeholders involved in the negotiation process need to be more involved and advocate intensively for the successful implementation of chapters' recommendations and closing them. Structural dialogue among decision makers and relevant stakeholders from the agricultural sector needs to be intensified. |
| P.6.1. The capacity of relevant subjects in the agricultural sector in the above-mentioned countries is poor in terms of applying for adequate funds that support agricultural research and development. P.6.2. The countries of the Western Balkans and their responsible government offices, agencies, and ministries for the agricultural sector do not recognise nor involve IPM measures and biocontrol in relevant national strategies and action plans; therefore, funding of the same is intensively lacking. P.6.3. Taking into account that IMP and biocontrol measures alike are weakly supported and implemented in the above-mentioned WB countries, their contribution to achieving sustainable development goals of the UN (SDGs), especially 2, 12, 13, 15, and 17, is also weak and invisible. | S.6.1. Relevant country government offices and ministries for international cooperation, EU accession, and agricultural questions need to intensify capacity building of relevant subjects for applying to projects at EU programmatic schemes. S.6.2. Biocontrol practitioners need to advocate intensively to decision makers on the benefits of the usage of IPM regulatory practices and biocontrol. S.6.3. An SDG achievement study with which IPM and biocontrol in agriculture are contributing (so far) and an assessment of their potential needs to be conducted. From this, a clearer picture of IPM presence in the agricultural sector in WB countries will be visible, and, therefore, action plans for achieving the mentioned SDGs could be set. |

* This designation is without prejudice to positions on status and is in line with UNSCR 1244/1999 and the ICJ Opinion on the Kosovo declaration of independence.

**Supplementary Materials:** The following are available online at https://www.mdpi.com/article/10.3390/agronomy14040668/s1, Table S1: Most frequent predators and parasitoids of *C. pyri* registered by recent inventory research in the WB region and European countries with similar agro-ecological conditions and a degree of predation/parasitism.

**Author Contributions:** Conceptualization, J.K. and A.I.; paper structure and methodology, A.I. and A.S.; writing—original draft preparation, A.I., J.K. and B.T.; writing—review and editing, B.T., B.K., T.B., K.P., I.S. and A.S.; visualization, B.T.; project administration, I.S. and K.P.; funding acquisition, A.S. and I.S. All authors have read and agreed to the published version of the manuscript.

**Funding:** This research was funded by the HORIZON2020 project "EU-wide farm network demonstrating and promoting cost-effective IPM strategies", with grant agreement ID: 101000339.



**Data Availability Statement:** Not applicable.

**Acknowledgments:** The authors would like to acknowledge limited usage of DeepL tool, an AI-based application, during the initial drafting phase of our manuscript for enhancing the English language presentation of the Abstract and a slight portion of the introductory section of our paper.

**Conflicts of Interest:** The author Boris Kuzmanovic is an employee of MDPI; however, he did not work for the journal Agronomy at the time of submission and publication.

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
