# Peer review of "Natural Enemies of the Pear Psylla, Cacopsylla pyri (Hemiptera: Psyllidae), and the Possibilities for Its Biological Control: A Case Study Review in the Western Balkan Countries"

_agronomy, doi:10.3390/agronomy14040668_

Round 1
Reviewer 1 Report
Comments and Suggestions for Authors
This manuscript reviews the biological data of available natural enemies of the pear psylla, Cacopsylla pyri, and the possibilities for its biological control in the Western Balkan countries. Due to C. pyri's rapid development of resistance to chemical treatments, biological control using natural enemies seems promising. Given that there is an abundance of research on the natural predators of C. pyri and their biocontrol applications across the globe, a review that synthesizes data of natural enemies against C. pyri already being used is crucial for countries who are planning to utilize natural enemies. This manuscript seems to have filled that gap, by providing detailed description of various natural enemies and an evaluation on their potential use in the biological control of the pear psylla, at least for Western Balkan countries. However, Although the authors summarized the information of various natural enemies, and gave an emphasis to the key natural enemies, but it is difficult to read from the text, if not impossible, exactly how many natural enemy species are available and what they are. It would be valuable to provide a table (or tables) to list out all the natural enemies of the pear psylla, including the species name, important biological information (such as generalist or specialist, which stage of the host it attacks), contribution in biocontrol, distribution, literature cited, etc. This table could be as supplemental material.
There are few minor comments.
1. Figure 1, change legend “Image of the pear psylla” to “Adult of the pear psylla”.
2. Figure 2, a better quality image should be used.
3. Figure 3, the image is too dark.
4. there are a few typos that the authors might want to go through the text carefully.
I don’t see any inappropriate references.
Comments on the Quality of English Language
Well-written, but there are a few typos that the authors might want to go through the text carefully.
Author Response
Author responses on all reviewer points have been uploaded as a document.

Reviewer 2 Report
Comments and Suggestions for Authors
Natural enemies of pear psyllids review
This paper contains a literature review of biological control of pear psyllid (Cacopsylla pyri) in the Balkan region. It provides a useful baseline for future biological control efforts in Europe.
The English usage is good. I did not find many things that required editing.
The main question I have is about biological control in a perennial crop if the main source of damage is a vectored pathogen. Biological control will always fluctuate, sometimes in favor of the parasites and predators, and sometimes in favor of the herbivore. If a vectored pathogen is present, and the crop is perennial, each fluctuation in favor of the herbivore allows unhindered spread of the pathogen, and potential demise of the perennial crop. I suggest looking at the literature about Diaphorina citri and huanglongbing for a similar situation. The only place that biological control seemed to manage HLB was in Réunion Island, and that situation has not been revisited for decades to see if the parasitoids continue to keep disease within economically viable limits. It is not working in neighboring Mauritius. In every other situation, including Florida, biological control was insufficient to manage disease. If you have other means of controlling the pathogen, resistant cultivars, for example, then biological control has a better chance of working.
Aside from the issue of the pathogens, it could be instructive to look at the D. citri/HLB system for other biological control information, such as the effects of predators versus parasitoids, effectiveness of pathogenic fungi, etc. There has been a lot of research on these topics, even though they do not control spread of HLB sufficiently.
Another general comment is that I had some difficulty following the flow of this article. One suggestion would be to make more use of subheadings. For example, the paragraph on mirids (page 9) follows a paragraph on beetles without an introduction. Another good addition would be a table listing all the organisms, organized by taxon, with complete scientific names and authors for reference for the reader. The taxonomic organization should be in the same order as the various taxa (beetles, mirids, spiders, mites, etc.) are presented in the paper.
Some specific comments:
1. Do you want to use “psylla” or “psyllid?” Please be consistent whichever you choose.
2. Page 4, bottom. Do bedbugs (Cimicidae) really prey on psyllids? If so, please elaborate.
3. Page 5, middle. A natural enemy that’s a good searcher is optimal. Get it there early in the season. Killing a few in early spring when the population is small is equivalent to killing an enormous number later in the season.
5. Page 11, last three lines. I think you mean that S. ariantes was recorded for the first time as a parasitoid of C. pyri.
Page 13, second paragraph. Are insecticides toxic to insect pathogenic fungi?
Page 13, general. Has Hirsutella been tried, or is it too difficult to rear? I suggest looking at the D. citri literature for more on psyllid pathogenic fungi.
Page 14. Kaolin has worked reasonably well for D. citri.
Pages 16-20. I think this table belongs in the supplemental material, but that is the choice of the editor. It seems tangential to the rest of the paper to me.
I ticked "accept after minor revision," but please do deal with the vectored pathogen issue. It is mentioned but not discussed. The presence of a vectored pathogen is a major issue for efficacy of classical biological control, particularly in a perennial crop, where loss of the crop can set production back by years.
Comments on the Quality of English Language
Only minor editing is needed.
Author Response

(The authors gave the same response as above.)

Reviewer 3 Report
Comments and Suggestions for Authors
Excellent contribution, uniquely I raise the doubt about whether to call juvenile stages larvae, being hemimetabolous organisms the psyllids present in juvenile stages nymphs.
It would just be that detail, I leave it to consideration.
Congratulations
Author Response

(The authors gave the same response as above.)
